

# Efficient Online Source Identification Algorithm for Integration within Contamination Event Management System

Jochen Deuerlein[1], Lea Meyer-Harries[1], Nicolai Guth[1]

[1]3S Consult GmbH, Karlsruhe, 76137, Germany

*Correspondence to*: Jochen Deuerlein (deuerlein@3sconsult.de)

**Abstract.** The automatic identification of the source of a contamination is an important component of an early warning and event management system for security enhancement of water supply networks. Whilst a number of algorithms have been published on the algorithmic development, only few information exists about the integration within a comprehensive real-time Event Detection and Management System. In the following the analytical solution and the software implementation of a
real-time source identification module and its integration within a web-based Event Management System is described. The development was part of the project SAFEWATER, which was funded under FP 7 of the European Commission.

## Introduction

For more than one decade a number of researchers have been working on methods for civil protection, real-time detection of contaminations and specific sensor development. Different software tools have been developed tackling problems such as
optimal placement of sensors in the system (e.g. TEVA-SPOT, 2016) and detection algorithms (e.g. CANARY, 2016). In the SAFEWATER project, which was funded by the European Union, a comprehensive water supply system security solution was developed. One part of the project was concerned with the development of new sensors for detection of chemical, biological and radio-nuclear contaminations. The other part dealt with development of a comprehensive Event Management Software (EMS) that collects all information from the field and from different software components that are
connected with the EMS including a newly developed Event Detection System (EDS) as well as offline and online hydraulic and water quality simulators. For response and mitigation of contamination events a software component for the identification of possible contamination sources has been developed, which was also integrated within the Web-GIS-based Event Management System of SAFEWATER. The communication channels between the individual modules were implemented by use of ActiveMQ (2015). The module enables the user to observe the current monitoring state of the
system (area observed by the sensors) also in case of no alarm. In case of an event the possible locations for the contamination sources are calculated and highlighted. A look ahead-calculation shows the estimated future spread of contaminant and indicates the valves that have to be closed for isolation of the contamination. All the calculations run automatically in regular time intervals (e.g. 1 min) in combination with the hydraulic real-time simulation. To guarantee the proper order of calculations a Petri-Net has been implemented within the online client SirOPC. The paper first gives an


Open Access · Drinking Water
Engineering and Science
Discussions

overview of the theoretical background of the algorithmic development followed by an outline of the integration within the EMS framework. At the end results of the application cases that were tested in SAFEWATER (mainly in the test lab at Water Supply Zurich) are presented and discussed.

## 2 Theoretical Background

### 2.1 Transport Model

The theory of modelling reaction and transport of substances within water distribution systems has been widely studied and described in a number of books and articles. A good overview of the methods that are included in most of the commercially available simulation software tools can be found in [7]. In the following a very brief presentation of the problem of contaminant injection with sharp quality front (discontinuity) is given. Reaction and diffusion are not considered in this case.

The related so-called Riemann problem is considered (page 49 in [8]):

$$(\text{PDE}) \qquad c_t + v c_x = 0 \qquad -\infty < \text{x} < \infty, \ \text{t} > 0 \tag{1a}$$

$$(\text{IC}) \qquad \text{c}(x,0) = c_0(x) = \begin{cases} c_L \ if \ x < 0 \\ c_R \ if \ x > 0 \end{cases} \tag{1b}$$

The initial value problem of Eq. (1) distinguishes from conventional transport equations only in the initial condition. Here, there are two constant water qualities $c_L$ and $c_R$ that are upstream and downstream of point $x = 0$. For $x = 0$ the initial

conditions have a discontinuity. For solution a Method Of Characteristics (MOC) can be used. When time passes the discontinuity is propagated along the characteristic lines with flow velocity $v$. Accordingly, the solution of this simplest case of Riemann problems, the solution of Eq. (1) is:

$$\text{c}(x,t) = c_0(x - vt) = \begin{cases} c_L \ if \ x - vt < 0 \\ c_R \ if \ x - vt > 0 \end{cases} \tag{2}$$

For the Riemann equation the characteristic that passes through $x = 0$ is of special interest. It separates the (concentration)

surface above the x-t-plane into one part where the concentration is $c_L$ and one part where the concentration is $c_R$. This particular characteristic line is the only one across which the concentration changes. In the implementation the IVP (1) is solved for all the pipes using a MOC. For later processing of the results the traces of a particle that travels along the special characteristic line that starts from $x = 0$ is stored. In the easiest case the flow velocity is constant and only the slope has to be stored. However, in extended period simulations the flow velocity and even the flow direction may change from time step

to time step. Therefore, it proved to be convenient to store the [time, position, value]-triples of all particles for every external time step. Since the flow velocities change only after such an external time step the full information [value, time, position] can be re-established from this information for any time and location.

In order to move from the pipe level - with the initial assumption of infinite pipe length from above - to the network level additional boundary conditions have to be considered for finite pipe length. The IVP then is transferred to an IBVP (Initial

Boundary Value Problem). In general, the combination of pipes at a junction requires the additional formulation of mixing



conditions at network nodes. Since no critical concentration can be given for the unknown substance, here, the binary information (contaminated yes/no) of the front is transferred unchanged to all pipes having outflows from the junction.

## 2.2 Source Identification

The Source Identification (SI) module distinguishes from existing solutions by its real-time capabilities, the integration within the EMS and the permanent calculation of the monitoring state of the sensor network even in the case without alarm. In the context of SAFEWATER one basic requirement was that the source identification algorithm must deliver results almost in real-time. For that reason all kind of optimisation based approaches were excluded because of the huge number of simulations that are required for stochastic optimisation models. Therefore, a more direct approach was chosen for the solution of the inverse problem. It has some similarities to the particle backtracking algorithm that was first presented by Shang et.al. (2002). Its core component is an event driven simplified transport algorithm that does not consider reaction and diffusion mechanisms. For solution a method of characteristics (MOC) is used.

One of the key performance indicators is the usage of memory. In the presented approach a special format was developed that is used for storing the characteristic lines of the transport equation MOC. This information can then be used for particle tracking (forward and backward) as well as reconstruction of time curves (at a certain location) or the concentration along a pipe at a certain time. In combination with the event driven method, which means that instead of time driven simulations only changes in water quality at the boundaries are considered, the memory requirements are minimized. An event is triggered every time a change in the boundary conditions occurs (for example start of intrusion in the forward case or release of a sensor alarm in the backward case). The algorithm sends a separator front through the system following the flow velocity of the water in the pipes (forward) or working against it (backtracking case).

For solution of the source identification problem at each external time step the alarm states of the sensors (positive, negative, unknown) are sent through the system in reverse time. If at least one of the sensors is positive the nodes and pipes upstream of the sensor are identified, which are possible candidates for the location of the contamination source. If more sensors have positive alarm states, normally released at different times, the results of the backtracking calculations are combined. A match is reached for nodes that are upstream of all positive sensors if the signals that are created by the backtracking procedure are consistent for all sensors. A specific weighting function has been developed that identifies the most probable locations for the contamination source based on the results of the backtracking method. Please note that also negative sensor alarms deliver important information. The backtracking of negative alarm states (under the assumption of perfect sensors) identifies the nodes upstream of these sensors and times when the signal passes the nodes. A contamination starting from such node before the calculated arrival time of the negative separator front is not possible since, in this case, the sensor alarm state must have been positive as well.

Another important application of backtracking of negative sensor alarms consists in the continuously updated monitoring state of the system. The backtracking algorithm calculates a kind of reverse water age for each location under the protection

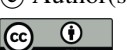



of the sensor network. That means that for any location the time of the last observation is known. In addition, the nodes and pipes that are not covered by the sensor network can be visualised and updated in real-time.

The simplified forward transport calculation is used for estimation of the future spread of contamination and the identification of isolation valves. Here, the assumption is again that more complicated water quality issues such as incomplete mixing, reaction and diffusion are of secondary importance for this particular problem because at the beginning of a real contamination event it is expected that the kind and severity of the substance is unknown. In the project SAFEWATER a second water quality solver was implemented that deals with all of these issues. Once more information about the chemical properties of the agent is available it can be used for more detailed calculation of concentrations.

## 3 Outline of the integration within the EMS

The Source Identification algorithm has been integrated within the OPC client software tool SirOPC. OPC (OPC Foundation, 2016) is a standard protocol for data communication in the automatisation industry. In SAFEWATER the software was extended by implementation of additional plugins for data exchange between the EMS, the simulator and the SI-application. Originally, SirOPC was developed for connecting hydraulic simulation software with common OPC Server software provided by SCADA systems for receiving and sending real-time operational data. Using the plugin technique of SirOPC the Source Identification plugin and the online hydraulic simulator were integrated within the SAFEWATER EMS. For communication with the EMS an ActiveMQ (2015) data adapter plugin has been developed in the project. The SirOPC ActiveMQ Plugin is able to receive messages about changes in alarm states of the sensors. Based on this information the online variables are updated in the SirOPC online data section from where the information is transferred to the SI-algorithm. By implementation, the developed SI-module runs in combined online mode together with the hydraulic solver. After each time step calculated by the hydraulic solver, the flow velocities in the Source Identification algorithm are updated and new backtracking runs (particle tracking in reverse time) are carried out. Positive alarms are generated by the EMS as soon as a pending alarm is acknowledged by the operator. After pressing the Acknowledge Alarm button in the EMS an ActiveMQ message is sent to SirOPC. The next SI-monitoring calculation considers the positive alarm and calculates the possible locations of a contaminant source that are consistent with the alarm. Based on a worst case assumption a unique source location is selected that serves as input for a simplified look-ahead transport calculation that gives an estimation over the future spread of contamination. As a possible response action the valves are identified that have to be closed for isolation of the contaminant. The results of the different algorithms are presented in a native GUI of the SI-application as well as in the EMS-Map. The execution loop, which is in mutual feedback with the hydraulic simulator, is controlled by so-called Custom Commands, a kind of virtual state machine (Petri-Net). The results of the SI-calculations are stored in a database that can be accessed by the EMS for further processing and visualisation.


# 4 Source Identification Module Example

## 4.1 Calculation of source candidates

The calculation of source candidates consists of the solution of the (IBVP), which is based on the simplified transport equation and the IVP in Eq. (1), in reverse time. For each sensor, the current alarm state is backtracked along the water
distribution network links until the signal reaches one of the water sources. During the backtracking process, the traces of a particle that travels with the signal through the system are stored. In other words, the particle PDE of Eq. (1) is solved for all the pipes using a MOC. The traces are the characteristic curves of the MOC.  It proved to be convenient to store the time, position, value triples of all particles for every external time step. Since the flow velocities change only once per external time step, the full information (value, time, position) can be re-established from this information. In the following section the
SI algorithm is demonstrated using the example of the test network in Zurich (Fig. 1).

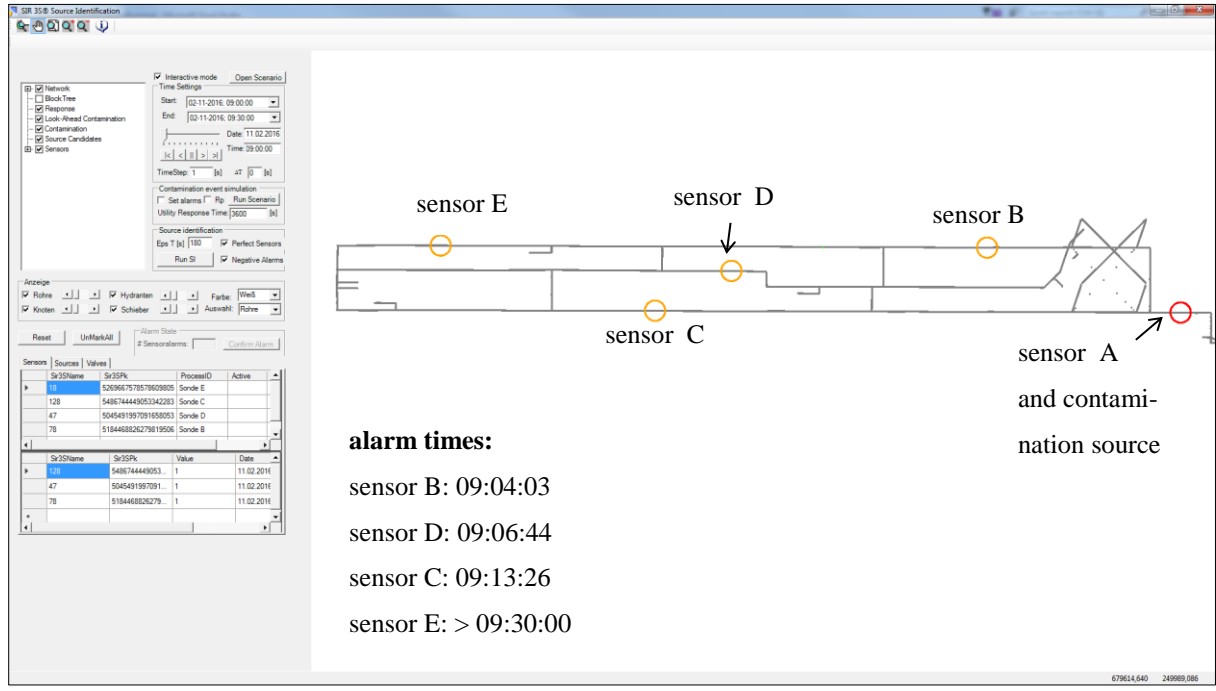

**Fig. 1: Test network with four sensors (orange circles) and contamination source (red circle)**

Sensors were installed at five locations. Four sensors are distributed over the network (orange circles) and one is located directly downstream the contamination source (red circle) in order to control the entrance water quality for the injection
scenarios. For demonstration purposes, the sensor alarm times were calculated by running a forward transport calculation that starts at 09:02 am. The SI-software tool is able to record the earliest time when the contamination passed the sensors. During the tests a valve on the lower left hand side was closed. Therefore, there is almost no flow in direction towards sensor E and the alarm time is beyond the scenario end time (09:30 am). The subsequent alarm times are also shown in Fig. 1.

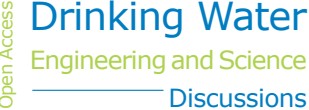



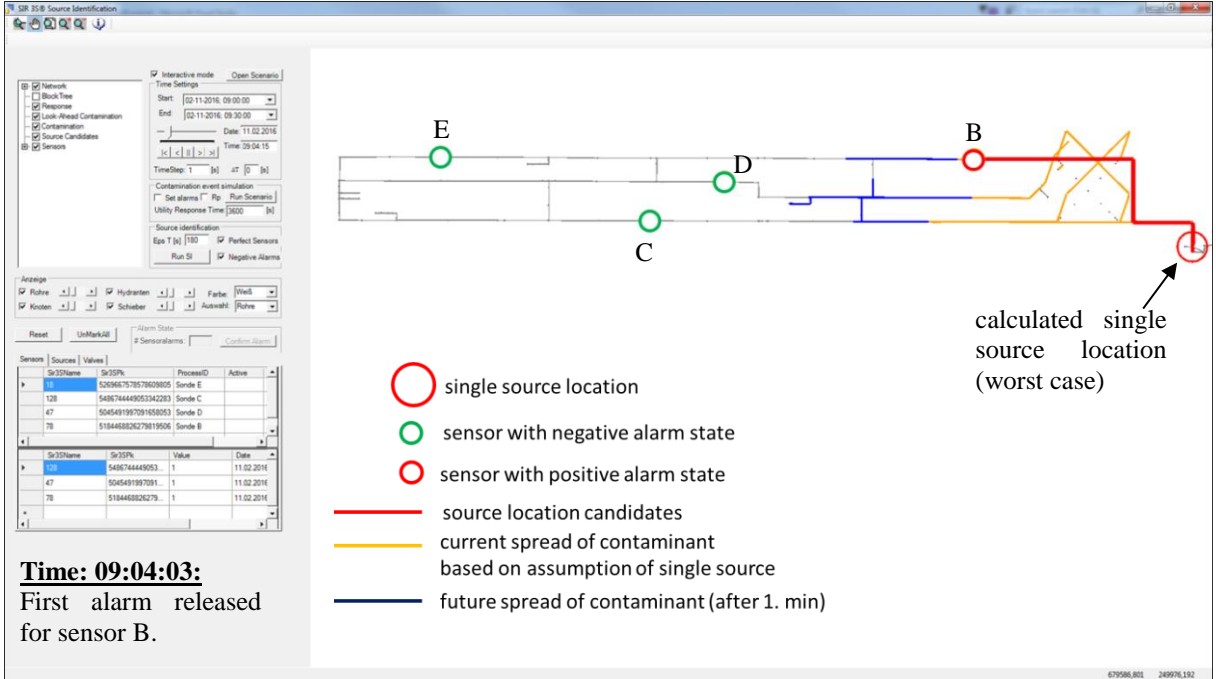

**Fig. 2: Release of first alarm at sensor B**

From Fig. 2 to Fig. 4 it can be seen that the more information is available (through subsequent additional alarms), the smaller

the area of possible source candidates (red marked pipes). In Fig. 2 all pipes upstream the sensor are possible locations. In Fig. 3 on the next page the combination of the upstream pipes of sensor B and of sensor D decreases the set of source candidates because the locations directly upstream of sensor B cannot explain the alarm at sensor location D. The area is further reduced after the third alarm (Fig. 4). Possible source locations must be also upstream of sensor C. The orange and blue marked pipes show the estimated spread at the current time and in a predefined look-ahead time, respectively.

**4.2 Impact of negative sensor alarms on source identification results and selection of single source**

The sensors do not only deliver valuable information in case of an alarm. Backtracking of negative sensor alarm states also helps to reduce the area of possible source candidates (see for comparison De Sanctis et. al., 2010). Assuming perfect sensors, the upstream locations of the negative sensors cannot be the contamination source since otherwise the contamination had to be detected by the sensor. Negative sensors can clear parts of the source candidate locations. A node (or pipe) is a

possible candidate for contamination source if the sensor signals of all sensors with positive alarms are observed and no signal of negative sensors can be found for these locations. A particular weighting schema has been implemented, considering also the duration and time delay between the different signals. The criterion for a node to be source candidate location is that all signals that passed that node are positive signals. The set of source candidate locations consists of a more



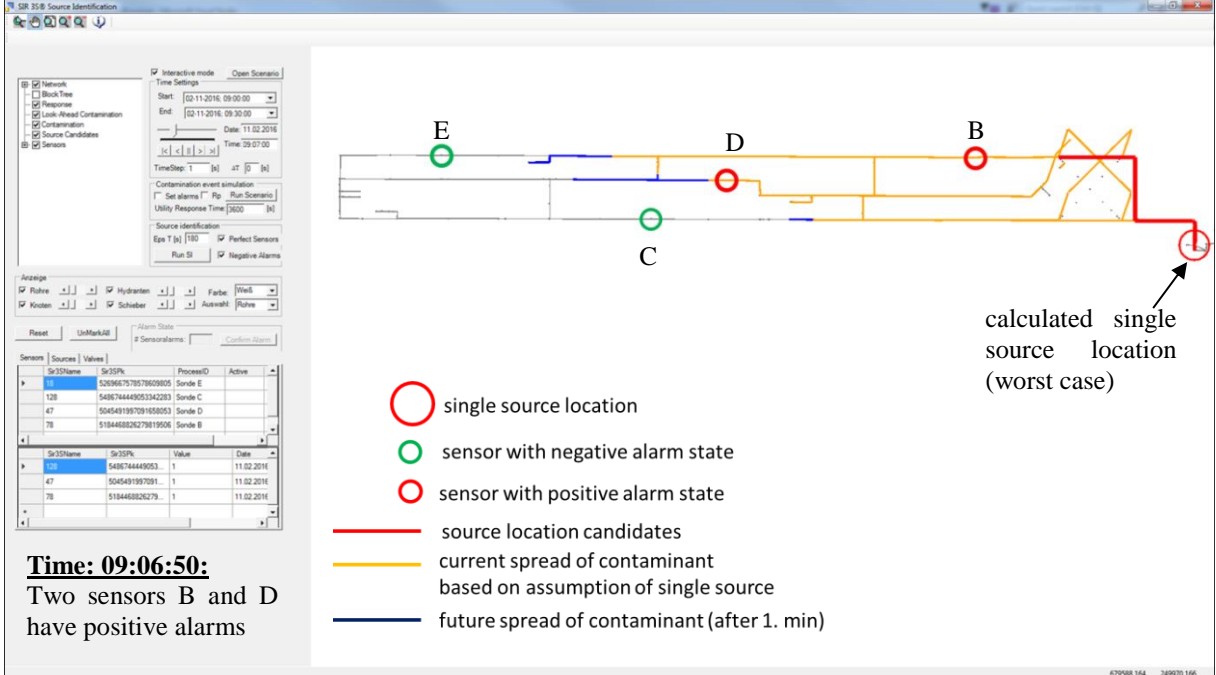

**Fig. 3: Release of second alarm at sensor D**

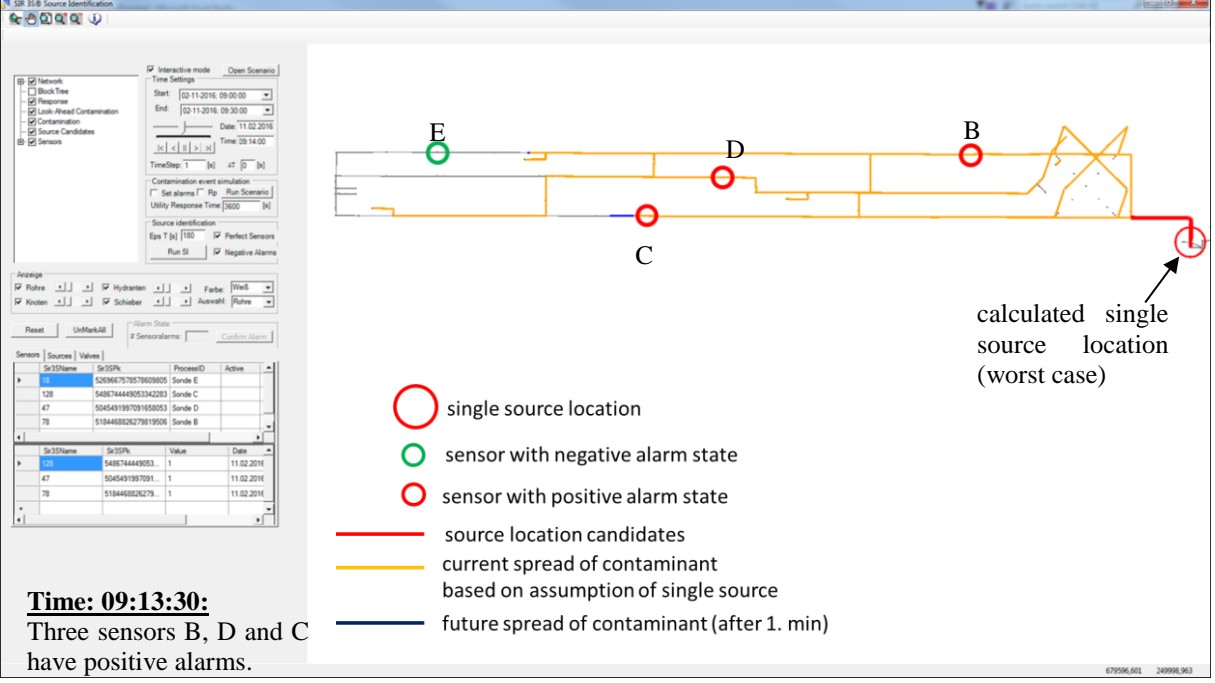

**Fig. 4: Release of third alarm at sensor C**



or less large area, depending on the design of the sensor network, the number of sensors and the topology of the network graph. For example, if forward calculations shall be carried out for the estimation of the current spread of contamination a single source node has to be selected. For the SAFEWATER SI-application, a worst-case selection algorithm was implemented, identifying the node with the biggest outflow among the source candidate locations.

## 5 Conclusion

In case of a contamination event, finding fast, efficient and simple response actions after detection is the key issue for mitigating contamination of drinking water distribution networks. The online source identification module indicates the possible locations of the source of contaminant and shows the current monitoring state at any location and any time. The module was integrated in combination with hydraulic online simulation within the SAFEWATER EMS. On the input side the hydraulic online simulator delivers system wide flow velocities needed for transport calculations based on actual process data that are received from a SCADA system using OPC technology. The alarm states of sensors are sent by the EMS using the communication channel ActiveMQ. As output the module generates current monitoring states of the network and in case of positive sensor alarm the source candidate locations. The output is visualised in the GIS map of the EMS. The functionality and applicability of the development of the approaches has been proven within the project for three pilot zone systems of real existing networks and a testnetwork on lab scale. Future work should focus on the enhancement of the calculation cycle. While the run time of the algorithms is considered to be sufficient also for large networks the huge amount of data to be processed and transferred through the different modules is still a challenge for large real world applications.

### Acknowledgement

This project has received funding from the European Union's Seventh Framework Programme for research, technological development and demonstration under grant agreement no. 312764.

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
