# Peer review of "Efficient Online Source Identification Algorithm for Integration within Contamination Event Management System"

_Drinking Water Engineering and Science, 2017_

## Referee Comment (RC1) · Anonymous Referee #1 · 29 Apr 2017

General Comments:

The manuscript "Efficient Online Source Identification Algorithm for Integration within Contamination Event Management System" describes an algorithm for the identification of the location of a contamination event in a water distribution system. I think the paper is interesting and should be published, because (as highlighted by the Authors) existing software and approaches are not implemented for a real-time (or near-real time) control. Therefore, this manuscript provides a clear contribution to the current knowledge. I think the paper is in general well written and I don't have any major issue. However, I think that the paper could be clarified in some aspects (see below). While I understand that the Authors may have been limited (in terms of paper length) by the

conference guidelines, I hope that the Journal will allow them to extend the manuscript, so that the content can be easily understood by a larger audience.

Specific Comments:

- Introduction, Line 27-28: I am not sure I understand this sentence. Do you mean that the algorithm is always running? Or it will be run (with a 1 minute time step) when there is an event?

- Introduction, Line 28-29: I don't understand why there could be problems with the order of calculations. Could you please explain it? (Are the calculations performed on multiple processors or something similar? Or is it in case some of the inputs from the network are delayed?) Or maybe you can move this line in page 4 (about line 29) and give some details here.

- Page 2, line 25: please, explain or give an example of external time step. I think this refers to the time step of the steady state hydraulic simulation (in contrast with the time step used in the method of characteristics), but please, confirm.

- Page 3, line 5: why are the computations performed even in case of no alarm? Maybe you can just refer the reader to section 4.2 (at least the reader will know that it will be explained somewhere).

- Page 3, line 15-17: "In combination with the event driven method, which means that instead of time driven simulations only changes in water quality at the boundaries are considered, the memory requirements are minimized." I don't understand what this sentence means. Could you please reword it? In particular, I don't understand the difference with time-driven simulation and event driven simulation.

- Page 3: line 25-26: "A specific weighting function has been developed that identifies the most probable locations for the contamination source based on the results of the backtracking method." Could you please give a bit more detail and explain why you need a weighting function? Is it to take into account possible sensor fail-

ures/malfunctioning or the fact that the concentration may be very small and not detected in some pipes/sensors?

- Page 4 lines 21-22: "Positive alarms are generated by the EMS as soon as a pending alarm is acknowledged by the operator. After pressing the Acknowledge Alarm button [. . .]". I think the software developed has the capability to automatically generate an alarm and start the computations. Maybe, at this stage, this is not performed in order to avoid several false alarms (or other reasons). However, I think it could be good if you could highlight that the operator response could be avoided.

- Page 4, Line 24: could you please give more details about the choice of the worst case assumption? How is the single contamination source chosen? Is it the one that could affect the largest number of users (or the users with the largest demands or something else?). I think this is related to line 5 of page 5: does the backtracking algorithm always reach a water source? I think this depends on the location of the sensors and the contamination event: for example, if the contamination is after the source and you have a 'negative' sensor next to the source, the algorithm will select a pipe/node upstream of a 'positive' sensor, but not the water source. I think you should clarify it in the manuscript. (I think you had this in page 8 lines 3-4, but I think it should be written (also) earlier in the paper).

- Fig 1: maybe you could add some information about the test network (e.g. total pipe length), in order to give an idea of how big the network was. Also, you may be able to show in the figure the location of the closed valve (which is described in line 18 of page 5).

- Page 8, line 10: Could you briefly discuss how these velocities are obtained? For example, are they computed from the hydraulic model based on some assumption on the demands? If so, this would introduce some uncertainty in the location of the contamination and the contamination zone. I understand that this is not the focus of this paper, but I think it should be mentioned. If the velocities are estimated/calculated

in a different way, please, provide some more details (e.g. do you have a real-time demand estimation algorithm?).

Technical Corrections:

- Please, check the references: in page 2, they are referred as [7] and [8] instead of using the Authors' names. Also, I am not sure these references are reported at the end of the manuscript. - Please, add Shang et al 2012 to the references. - Eq 1a and 1b: please, define PDE, IC and the symbols in the equation (ct, cx, and c0) - Please, define IVP (Initial Value Problem) in line 13 of page 2. - Page 8, line 16: I think the sentence should read "While the run time [. . .] is considered to be sufficiently short also for large networks, [. . .]"

---

## Author Comment (AC1) · 5 May 2017

please find attached response to review and modified discussion paper

———————————————————

[Figure]

**Efficient Online Source Identification Algorithm for Integration within Contamination Event Management System**

Jochen Deuerlein[1], Lea Meyer-Harries[1], Nicolai Guth[1]

[1]3S Consult GmbH, Karlsruhe, 76137, Germany

5 *Correspondence to*: Jochen Deuerlein (deuerlein@3sconsult.de)

**Abstract.** The automatic identification of the source of a contamination is an important component of an early warning and event management system for security enhancement of water supply networks. Whilst a number of algorithms have been published on the algorithmic development, only few information exists about the integration within a comprehensive real-time Event Detection and Management System. In the following the analytical solution and the software implementation of a

10 real-time source identification module and its integration within a web-based Event Management System is described. The development was part of the project SAFEWATER, which was funded under FP 7 of the European Commission.

**Introduction**

For more than one decade a number of researchers have been working on methods for civil protection, real-time detection of contaminations and specific sensor development. Different software tools have been developed tackling problems such as

15 optimal placement of sensors in the system (e.g. TEVA-SPOT, 2016) and detection algorithms (e.g. CANARY, 2016). In the SAFEWATER project, which was funded by the European Union, a comprehensive water supply system security solution was developed. One part of the project was concerned with the development of new sensors for detection of chemical, biological and radio-nuclear contaminations. The other part dealt with development of a comprehensive Event Management Software (EMS) that collects all information from the field and from different software components that are

20 connected with the EMS including a newly developed Event Detection System (EDS) as well as offline and online hydraulic and water quality simulators. For response and mitigation of contamination events a software component for the identification of possible contamination sources has been developed, which was also integrated within the Web-GIS-based Event Management System of SAFEWATER. The communication channels between the individual modules were implemented by use of ActiveMQ (2015). With the help of continuous calculations the module enables the user to observe

25 the current monitoring state of the system (area observed by the sensors) also in case of no alarm. In case of an event the possible locations for the contamination sources are additionally calculated and highlighted. A look ahead-calculation shows the estimated future spread of contaminant and indicates the valves that have to be closed for isolation of the contamination. All calculations run automatically in regular time intervals (e.g. 1 min) in combination with the hydraulic real-time simulation. To guarantee the proper order of actions and calculations a Petri-Net has been implemented within the online

**Fig. 1.**

Drink. Water Eng. Sci. Discuss.,
doi:10.5194/dwes-2017-16-RC1, 2017
The manuscript "Efficient Online Source Identification Algorithm for Integration within Contamination Event Management System" describes an algorithm for the identification of the location of a contamination event in a water distribution system. I think the paper is interesting and should be published, because (as highlighted by the Authors) existing software and approaches are not implemented for a real-time (or near-real time) control. Therefore, this manuscript provides a clear contribution to the current knowledge. I think the paper is in general well written and I don't have any major issue. However, I think that the paper could be clarified in some aspects (see below). While I understand that the Authors may have been limited (in terms of paper length) by the conference guidelines, I hope that the Journal will allow them to extend the manuscript, so that the content can be easily understood by a larger audience.

Specific Comments:

- Introduction, Line 27-28: I am not sure I understand this sentence. Do you mean that the algorithm is always running? Or it will be run (with a 1 minute time step) when there is an event?

The backtracking algorithm calculates the propagation of sensor signals in reverse time for both, negative (no alarm) as well as positive (alarm) signals. As result, the monitoring state of the observed network parts is always available. In other words for each location the time and the value of the last observation are regularly updated.

- Introduction, Line 28-29: I don't understand why there could be problems with the order of calculations. Could you please explain it? (Are the calculations performed on multiple processors or something similar? Or is it in case some of the inputs from the network are delayed?) Or maybe you can move this line in page 4 (about line 29) and give some details here.

The monitoring system in combination with hydraulic online simulation consists of several software components that share data using customized interfaces. In order to maintain proper workflow, the sequence of calculations follows specific rules. For example, the

**Fig. 2.**
[Figure]

---

## Referee Comment (RC2) · Anonymous Referee #2 · 9 May 2017

Dear Editor

the paper proposes an algorithm for the identification of the source of a contamination and its integration within a real-time management system. The paper deals with a hot topic in the field of water distribution system and so it is interesting for operators, also because it presents a novel software that can be very useful for the water utilities. However, the paper is not clear in some parts, but it can be improved and so published if the Authors will take into account some major and minor revisions.

Major Comments:

1) Introduction is poorly written with a limited state of art about the topic of water distri-

bution system protection from contamination. More than half of the abstract is devoted to the developed software. It is important to describe better the state of the art (both in terms of contamination risk and in terms of methodology to face the problem) and previous papers that deal with the problem of source identification. The following works can be related to the paper topic (in which several other papers can be found in the references):

a) Chang, N.-B., Pongsanone, N.P., Ernest, A. (2011). Comparisons between a rule-based expert system and optimization models for sensor deployment in a small drinking water network Expert Systems with Applications, 38 (8), pp. 10685-10695.

b) Hall, J., Zaffiro, A.D., Marx, R.B., Kefauver, P.C., Radha Krishnan, E., Haught, R.C., Herrmann, J.G. (2007). On-line water quality parameters as indicators of distribution system contamination Journal / American Water Works Association, 99 (1), pp. 66-77.

c) Di Nardo, A., Di Natale, M., Musmarra, D., Santonastaso, G.F., Tzatchkov, V., Alcocer-Yamanaka, V.H. (2015). Dual-use value of network partitioning for water system management and protection from malicious contamination, Journal of Hydroinformatics, 17 (3), pp. 361-376.

2) The description of the methodology and the model are confused (maybe because too "compressed") and some parameters are not explained – page 2, equation 1a (e.g. ct and cx). Are they refuse? Please check to make paper consistent and more clear. I suggest to use a flow chart to improve the methodology description.

3) Page 4, line 24: "based on a worst case assumption of a unique source location is selected ....", taking into account the different hypothesis and the simplified model used, is realistic to focus on a unique source location or it will be better to identify an area on which focus on the attention?

4) Page 5, line 9: "In the following section the SI algorithm is demonstrated using the example of the test network in Zurich (Fig. 1).". It should be interesting to provide

some information about this water distribution network, and above all, it should be more attractive to test the algorithm on a real system.

Minor revision:

1) Page 1, line 2: "Whilst a number of algorithms have been published on the algorithmic development . . .", please change.

2) Page 1: please pay attention to the sequence of tenses.

3) Pay attention to reference format in the text, (e.g. page 2, line 8 – ".... software tools can be found in [7]", while at page 3, line 10 "... page 6, line 12 "... of possible source candidates (see for comparison De Sanctis et. al., 2010)".

4) Pay attention to reference in the text; at page 2, line 8 is reported the seventh reference, where are the previous six? Please check. Furthermore, the references at the end of paper are not numbered.

5) Page 3, line 10: Shang et al, (2002), is not reported in the references list.

6) I suggest improving the English language.

7) I suggest improving the quality of figures.

8) Figures are very large; if the authors reduce their dimensions they can save space for better explain the methodology and the results.

---

## Author Comment (AC2) · 17 May 2017

Please find attached the full response to review and modified discussion paper. We are sorry for the mistake of uploading only the first pages.

Please also note the supplement to this comment: http://www.drink-water-eng-sci-discuss.net/dwes-2017-16/dwes-2017-16-AC2-supplement.zip
* * *

---

## Author Comment (AC3) · 1 Jun 2017

Please find attached: - response to Referee 2 - revised paper - revised abstract

Please also note the supplement to this comment:
http://www.drink-water-eng-sci-discuss.net/dwes-2017-16/dwes-2017-16-AC3-supplement.zip

---

## Author Comment (AC4) · 1 Jun 2017

Please keep in mind that the answers to Referee #1 may by outdated in some points as the response to Referee #2 followed afterwards.
* * *